# Comparing the Impact of Road Networks on COVID-19 Severity between Delta and Omicron Variants: A Study Based on Greater Sydney (Australia) Suburbs

**DOI:** 10.3390/ijerph19116551

**Published:** 2022-05-27

**Authors:** Shahadat Uddin, Haohui Lu, Arif Khan, Shakir Karim, Fangyu Zhou

**Affiliations:** School of Project Management, Faculty of Engineering, The University of Sydney, Forest Lodge, Sydney, NSW 2037, Australia; haohui.lu@sydney.edu.au (H.L.); arif.khan@sydney.edu.au (A.K.); shakir.karim@sydney.edu.au (S.K.); fangyu.zhou@sydney.edu.au (F.Z.)

**Keywords:** suburban road network, delta variant, omicron variant, network analysis, COVID-19 severity

## Abstract

The Omicron and Delta variants of COVID-19 have recently become the most dominant virus strains worldwide. A recent study on the Delta variant found that a suburban road network provides a reliable proxy for human mobility to explore COVID-19 severity. This study first examines the impact of road networks on COVID-19 severity for the Omicron variant using the infection and road connections data from Greater Sydney, Australia. We then compare the findings of this study with a recent study that used the infection data of the Delta variant for the same region. In analysing the road network, we used four centrality measures (degree, closeness, betweenness and eigenvector) and the coreness measure. We developed two multiple linear regression models for Delta and Omicron variants using the same set of independent and dependent variables. Only eigenvector is a statistically significant predictor for COVID-19 severity for the Omicron variant. On the other hand, both degree and eigenvector are statistically significant predictors for the Delta variant, as found in a recent study considered for comparison. We further found a statistical difference (*p* < 0.05) between the R-squared values for these two multiple linear regression models. Our findings point to an important difference in the transmission nature of Delta and Omicron variants, which could provide practical insights into understanding their infectious nature and developing appropriate control strategies accordingly.

## 1. Introduction

Two years after the first reported coronavirus case, the total number of confirmed COVID-19 cases has surpassed 260 million worldwide [1]. Recently, a road-network-based approach was utilized to analyze transmissibility [2]. In that study, COVID-19’s severity and each suburb’s vulnerability were assessed using the centrality measures from a network constructed with suburbs as nodes and roads between suburbs as edges. It was found that the degree centrality of this suburban road network was a strong and statistically significant predictor of both vulnerability and severity [2]. As a result, studying road networks can help researchers better comprehend Delta and Omicron’s infectivity and severity levels. However, there has been no comparison between the impact levels of this suburban road network on the two variants. Therefore, this study aims at bridging this gap by utilizing the same network for respective variants and tries to quantify the differences, if there are any.

Since the beginning of July 2021, newly confirmed COVID-19 cases have been more than three million per week globally, reaching a maximum of 23.25 million in mid-January 2022 and presently (May 2022) trending down to four million [3]. This indicates that both the original severe acute respiratory syndrome coronavirus 2 (SARS-CoV-2) and its many variants, including Delta and Omicron, are still being transmitted between different hosts and species on a big scale [4]. Although some scholars have anticipated that this pandemic will come to an end soon [5], it is still heavily concerning that SARS-CoV-2 and its various variants continue circulating widely throughout the global population, especially considering the more contagious nature of the most recent Delta and Omicron variants. This study concentrates on these two variants and the effects of the road networks on their transmission in the Greater Sydney, Australia region.

The Delta variant has caused a new substantial infection wave worldwide, jeopardizing the COVID-19 eradication effort and putting humanity’s destiny in trouble [1]. The Delta variant has spread to nearly every country since its first detection in October 2020 and it has emerged as the most common circulating variant in newly confirmed COVID-19 cases globally [6]. There is currently little evidence that the Delta variant’s transmission channels have changed considerably. However, it is evidenced critically that the Delta variant is far more transmissible than the original SARS-CoV-2 strain [7] and the Delta variant is predicted to be even 60% more transmissible than the Alpha variant, which was already highly infectious [8]. Researchers have also been trying to uncover the biological reasons why the Delta variant acts like this. It has been suggested that the Delta variant has significantly more viral particles in the airways of patients [9]. According to Li et al. [10], the mean oropharyngeal swab viral load of Delta-infected people was roughly 1260 times more than the cases infected by the original SARS-CoV-2. Bjorkman et al. [11] also found that viral load is strongly connected to SARS-CoV-2’s transmissibility.

The Omicron variant, on the other hand, has been labelled by the World Health Organization (WHO) as a super ‘variant of concern’ due to its infectious and vaccine-escape mutations. The spike protein, which is the principal antigenic target of antibodies produced by infections or vaccination, has 32 mutations in this variant [12]. Mutations determine the SARS-CoV-2 variant’s critical infectivity and antibody resistance in the spike (S) protein receptor-binding region. By presenting a comprehensive quantitative analysis of Omicron’s infectivity, vaccine breakthrough and antibody resistance, it was discovered that Omicron may be over ten times more contagious than the original virus, or about 2.8 times as infectious as the Delta variant, and that Omicron may have an 88% chance of evading current vaccines. [12]. Therefore, the arrival of the Omicron variant has caused significant concern and researchers have been trying to investigate the extent to which this new variety may jeopardize existing vaccines [13,14]. However, in general, fully characterizing the complete impact of Omicron’s S protein mutations on current vaccines in the world’s populations is nearly impossible due to various reasons, e.g., diverse immunological responses, different factors impacting the creation of antibodies and even the statistical models in different experimental settings.

Apart from the properties of the virus itself, other factors have also been studied to verify the transmissibility of the COVID-19 variants. It has been found that there exists a negative correlation between transmission and temperature or humidity, but a positive correlation between recovery and these two factors [15]. Some academics have also unveiled a U-shaped relationship between outdoor ultraviolet exposure and weak positive associations with air pressure, wind speed, precipitation, diurnal temperature, SO_2_, and ozone [16]. There is no correlation between infectious viral titres (IVTs) and age or sex [9]. Previously, we found that road networks impact COVID-19 severity and vulnerability for the Delta variant. In this study, we investigate the hypothesis for the Omicron variant and explore the difference in spread pattern between these two variants.

## 2. Materials and Methods

### 2.1. Data

For this study, we considered data from 19 local government areas (LGAs) of Greater Sydney in New South Wales, Australia. There are 137 postal areas in these 19 LGAs. Google Maps was used to extract road connections between suburbs. The infection data were taken from the Australian Bureau of Statistics (ABS) [17]. We considered infection data for five weeks for the Delta variant, starting from 16 June 2021. This study evaluated only one week of data for the Omicron variant, starting from 2 February 2022. The Delta cases were reported only through the polymerase chain reaction (PCR) test, administered at different healthcare facilities. Omicron cases were reported based on both PCR and home-based rapid antigen tests (RATs). There were inconsistencies in reporting the RAT outcomes during the first couple of weeks since its introduction for COVID-19 testing in late December 2021. After that, the State Government of New South Wales, Australia, made it mandatory to report any positive outcome based on at-home RATs [18], which made the reporting of RATs more consistent and reliable from late January 2022. Due to this, we considered one week of Omicron data starting from 2 February 2022. In our Omicron data, there is a small portion of Delta cases. As reported by the State Health Chief in mid-January 2022, more than 90% of the reported cases were the Omicron variant in our data collection area since January 2022 [19]. Since those cases are distributed uniformly across the suburbs considered in this study, the tiny presence of Delta cases should not bias any relevant Omicron findings. We performed all research methods by following the appropriate guidelines and regulations.

### 2.2. Construction and Analysis of Suburban Road Networks

Suburban road networks show interconnections between a set of suburbs. Each suburb is represented by a node. Connections between nodes are represented through edges. Postcodes are seen as nodes in the construction of the suburban road network, i.e., a postcode denotes a suburb. An edge between two suburbs shows that a route connects them directly without passing through any intermediary suburbs. We used the number of roads joining two suburbs as an edge weight. We used Google Maps and ABS data [17] to locate suburb boundaries and count the number of roads connecting two suburbs. Considering a portion of the data from our research dataset (i.e., for nine suburbs), Figure 1 illustrates a small segment of the entire suburban road network considered in this study. Although such a network can be based on all transportation media (e.g., road, bus and rail), this study considered only the road network due to the imposed lockdown on bus and rail services during the pandemic.

This study used four network centrality measures (degree, closeness, betweenness and eigenvector) and the coreness measure to analyse the suburban road network. These measures are briefly defined in Table 1.

### 2.3. COVID-19 Severity Measure

This measure indicates the prevalence statistics of COVID-19-infected patients. The COVID-19 severity is the number of infected patients per 1000 population for a given suburb. For a suburb with comparatively more COVID-19 cases, the value for this measure will be higher. Similarly, a lower value for this measure indicates that the underlying suburb has comparatively fewer COVID-19-infected patients.

### 2.4. Regression Model

Since this study compares the impact of road networks on COVID-19 severity between Delta and Omicron variants, we considered the same multiple linear regression (MLR) and random forest regression (RFR) models previously used by Uddin et al. [2] for the Delta variant. The following equation represents the basic model.
Severity=f(degree, closeness, betweenness, eigenvector, coreness)

Our study used both MLR and RFR models for the Delta and Omicron data. Herein, we refer to $MLR_{D}$ and $RFR_{D}$ for the corresponding regression models based on the Delta data. For the Omicron data, they are $MLR_{O}$ and $RFR_{O}$. Since we considered the same LGAs as Uddin et al. [4] and no new road was constructed since their study, the suburban road network remains unchanged. Hence, the numerical value for corresponding network measures resulting from this suburban network is unchanged across these four models. However, the dependent COVID-19 severity measure changes.

### 2.5. Comparing R-Squared Values

We use the same regression models for modelling Delta and Omicron infectious data. That means we need to follow an approach to compare R-squared values across two datasets. For this purpose, we followed the method proposed by Olkin and Finn [21]. According to their proposed method, we first need to determine the standard error (SE) for R-squared values of the underlying regression model using the following equation.
$$SE_{R_{i}^{2}}^{2}=\sqrt{\frac{4R^{2}\left(1-R^{2}\right){\left(n-k-1\right)}^{2}}{\left(n^{2}-1\right)\left(n+3\right)}}$$
where $SE_{R_{i}^{2}}^{2}$ is the squared of the SE for the R-squared value of the $i^{th}$ model, n is the sample size and k is the number of independent variables.

We then need to find the difference in the $SE_{R^{2}}$ values for $i^{th}$ and $j^{th}$ regression models using the following equation.
$$SE_{R_{i}^{2}-R_{j}^{2}}=\sqrt{SE_{R_{i}^{2}}^{2}+SE_{R_{j}^{2}}^{2}}$$

After that, to find the 95% confidence interval (CI) of this measure ($SE_{R_{i}^{2}-R_{j}^{2}}^{2}$) we need to multiply it by 2, since the corresponding t value for a 95% confidence is close to 2 (for df=120, it is 1.984). If the 95% CI includes 0, there is no statistically significant difference between the underlying regression models. An inclusion of 0 in the 95% CI of $SE_{R_{i}^{2}-R_{j}^{2}}^{2}$ indicates a chance that the confidence could be 0. Otherwise, there is a statistically significant difference between them.

## 3. Results

All experiments were performed using Python’s Sci-Kit library [22]. The road network for the entire study area of 137 suburbs is presented in Figure 2. We followed the approach described in Section 2.2 in constructing this aggregated network.

The multiple linear regression results for the COVID-19 severity measure are presented in Table 2. According to the p-values, eigenvector centrality is statistically significant at *p* < 0.05 for both Delta and Omicron variants. The magnitude of the *t*-value can be used to determine the relative importance of the independent variables. On the other hand, degree centrality is found to be statistically significant only for the Delta variant at *p* < 0.05. We noticed that the coefficients’ signs are the same in both variants.

We then compared the R-squared values across multiple linear regression and random forest models using Delta and Omicron. The results are presented in Table 3. The R-squared for the multiple linear regression for the Omicron variant is 0.040. We compared this R-squared with the R-squared value obtained using the Delta variant data. Since the approximated 95% confidence interval excludes 0, the difference between these two R-squared values is statistically significant at *p* ≤ 0.05. At the same time, the R-squared for the random forest regression is 0.363 for the Omicron variant. The 95% confidence interval for R-squared values between the Omicron and Delta variants also does not include 0. Therefore, the difference in the R-squared values between these two variants is significant at *p* ≤ 0.05.

The feature importance specifies which features are important in the underlying model and aids in a better understanding of the model. Figure 3 shows the feature importance for both variants. This figure is based on the random forest regression. Degree and eigenvector centrality measures rank first in the two scenarios, respectively, and both of them are much greater than the one that comes second.

## 4. Discussion

This study aimed to compare the severity of two COVID-19 variants—Delta and Omicron—from a network perspective. Extending our previous work on the Delta variant, here we modelled both variants using the same set of network features to find the model effectiveness and compare the importance of different features. The model performance and feature importance also helped us to approximate the underlying dynamics of Delta and Omicron transmission and people’s mobility during social restrictions.

Table 2 shows that the t-values from multiple linear regression models have the same positive or negative sign for each network measure across the Delta and Omicron variants. This broadly shows that the network features affect the severity of both variants similarly in the same direction. However, the model performance varied considerably between the variants. The R-squared values from Table 3 indicate that the random forest algorithm is better than multiple linear regression in model performance and capturing variability. COVID-19 transmission within a population is non-linear in nature. The different road connections with neighbouring suburbs make the network features non-uniform. As a result, the non-linear property of the random forest algorithm better captured the variability and performed better. In addition, we can see that the Delta variant had a better R-squared value than the Omicron variant. This is probably because of the greater transmissibility of the Omicron variant and the absence of a hard lockdown during the Omicron study period that led to more frequent human movement transcending neighbouring suburb boundaries. As a result, the network variables were not effective enough to capture the severity during Omicron compared to the earlier Delta outbreak period.

We also looked at the feature importance between the variants for multiple linear regression (Table 2) and random forest regression (Figure 3). The importance measures reveal some interesting perspectives. Summarily, we can say that degree centrality was a dominating measure for COVID-19 severity during the Delta variant. On the other hand, it is mainly eigenvector centrality (as seen in both the MLR and RF models), followed by betweenness centrality (seen in the RF model), for the Omicron variant. For the Delta variant, degree centrality is a significant variable in the MLR model and is positioned top in terms of feature importance in the RF model. Referring to Section 2.2 on how the road network is constructed, degree centrality corresponds to the number of roads that connect the suburb with its neighbouring suburbs. Thus, degree centrality stands as a proxy for human mobility across the suburbs during the Delta outbreak. Although the suburbs were in lockdown with only emergency movement allowed within a 10 km (later, 5 km) radius, people could still cross suburb boundaries, as most suburbs are smaller than that. Thus, we can assume that road connections from immediate neighbourhoods correspond to the incoming traffic, which correlated with infection rate. Consequently, degree centrality became the most important feature for the Delta variant.

On the other hand, it is interesting to see eigenvector centrality becoming the most important feature for the Omicron variant, instead of the more intuitive degree centrality. From the network perspective, degree centrality concerns the direct connections a suburb has with its immediate neighbours, while eigenvector centrality measures the influence within the network. Influence is recursively defined in such a way that neighbours of an influential node also achieve high scores and thus influence trickles down or diffuses from highly influential nodes to their neighbours, from these neighbours to their neighbours and so forth. In our road network perspective, a suburb having high eigenvector centrality means either it has higher road connectivity with neighbouring suburbs or it is located near suburbs having high road connectivity, or both. Now, suppose we question why, instead of the immediate neighbourhood’s degree centrality, the much broader network-position-based eigenvector centrality became a dominant network feature for the Omicron variant. In that case, we need to look at two potential reasons—transmissibility of the variant and restriction measures during that time. The Omicron variant is much more transmissible compared to the Delta variant. In separate studies from Denmark and Japan, researchers estimated that Omicron is 3.19 to 4.2 times more transmissible than Delta [23]. For the Omicron research period of early February 2022, Sydney did not have any lockdown or movement restrictions in place. Thus, we can assume that people’s movement was not restricted to immediate neighbouring suburbs. Instead, they might have moved multiple times for daily activities. As a result, Omicron’s severity risk factor was probably not confined within the immediate neighbourhood, but a strategic position within or near highly connected suburbs mattered more, as people’s mobility was unrestricted. That explains why Omicron had eigenvector centrality as the dominant feature. In addition, with similar reasoning (albeit with a lesser effect due to lockdown), we can see why eigenvector was the second most important feature after degree centrality for the Delta variant (Figure 3 from RF; MLR for the Delta variant also showed eigenvector centrality to be significant, as shown in Table 2).

This study considered only one-week infection data for the Omicron variant, since the COVID-19 cases remained relatively stable during the corresponding data collection period. Consideration of more Omicron data (e.g., four weeks instead of one week) could change the parameter values of the models we applied. However, it is improbable that the inclusion of any such additional Omicron data would make any changes to the model significance values.

We noticed a significant difference in the performance of regression models between the Delta and Omicron variants (Table 3). The R-squared values (for both MLR and RF models) based on the Delta data are higher than those from the Omicron data. This difference is statistically significant at *p* ≤ 0.05. As mentioned earlier, Omicron is much more transmissible than Delta. Such high transmissibility would lead to more Omicron infections among nearby people (e.g., living within the same suburb and next-door neighbours). This will eventually weaken our network measures in explaining the variability of COVID-19 severity across suburbs that have different values for network measures.

## 5. Conclusions

This study revealed how network features from suburban road networks vary in terms of importance between Delta and Omicron outbreak severity. Most interestingly, we observed that underlying factors, e.g., lockdown and variant transmissibility, affected the network dynamics. Depending on these, connectivity (degree centrality) or structural position (eigenvector centrality) dominated regression and random forest models. This could be very insightful in modelling any potential future pandemics. In addition, it would be interesting to include other modes of public transportation (e.g., rail network and bus transit network) in the dataset to better fit the model. For the Omicron variant especially, when there was no lockdown, the inclusion of other transportation modes into the network might improve the model’s performance.

## Figures and Tables

**Figure 1 ijerph-19-06551-f001:**
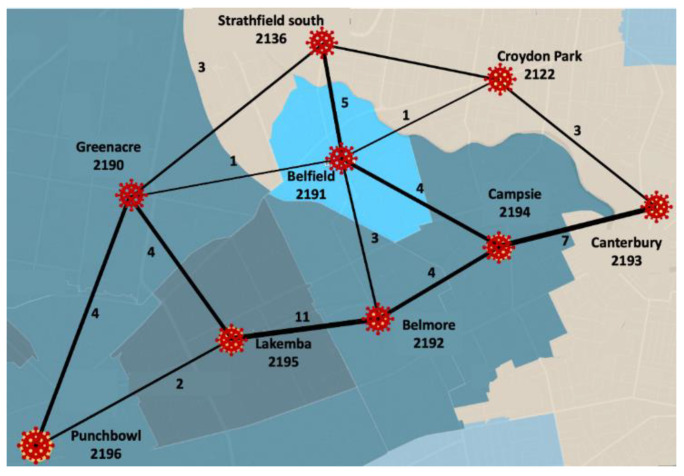
An example of the suburban road network among nine suburbs. This network is a small segment of the entire suburban road network considered in this study. The complete suburban network for the whole study region is presented in Figure 2.

**Figure 2 ijerph-19-06551-f002:**
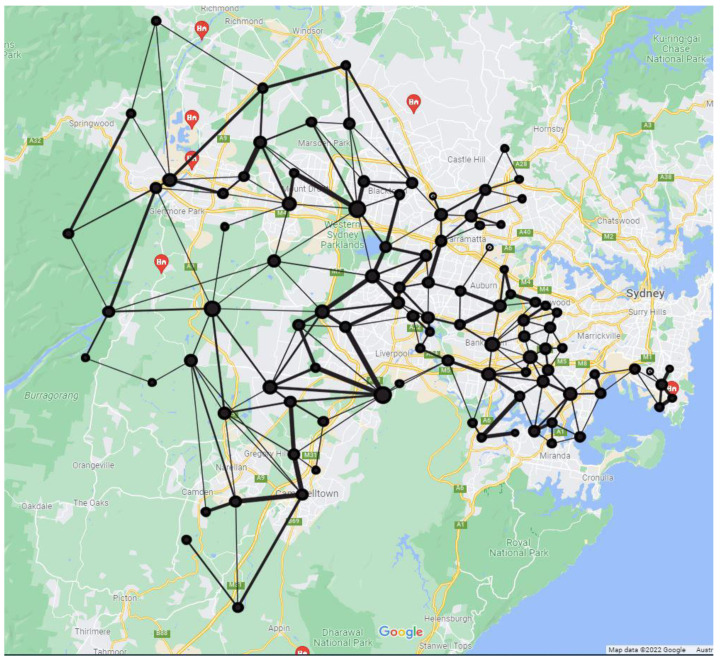
The suburban road network for the entire study area of 137 postal areas. The node size is set according to their degree centrality values. The edge weight between two suburbs is proportionate to the number of roads connecting them.

**Figure 3 ijerph-19-06551-f003:**
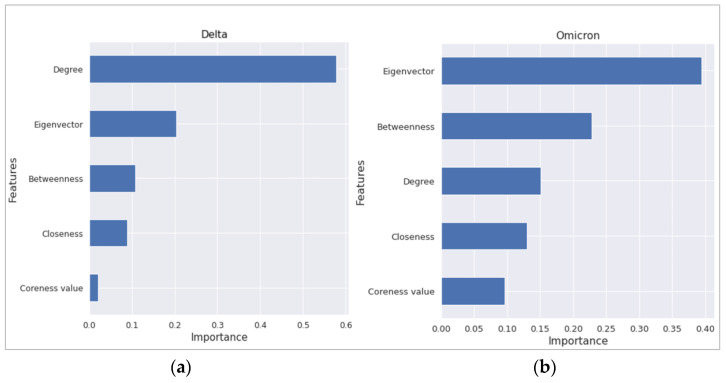
Feature importance results from the random forest regression for (**a**) severity for Delta variant and (**b**) severity for Omicron variant.

**Table 1 ijerph-19-06551-t001:** Definitions of network analysis measures.

Measurement	Definition
Degree centrality [20]	It indicates the number of edges that are incident to a node. Suburbs with a high degree have more connections to other suburbs and vice versa.
Closeness centrality [20]	It represents the inverse of the geodesic (or shortest) paths between a node and every other node in the network. This measure shows the ease of travelling between suburbs. Suburbs with a high closeness can have faster and easier access to adjacent suburbs.
Betweenness centrality [20]	It represents the number of other nodes that have to travel through a certain node in order to obtain their shortest path. Suburbs with a high betweenness centrality are in the shortest path of many other pairs of suburbs.
Eigenvector centrality [20]	It specifies the degree to which a node is linked to other significant nodes. A high eigenvector centrality for a node means that it is connected to many other nodes that also have a high score.
Core–periphery structure [20]	A coreness score is assigned to each node in the network. A node that is closely linked to other network nodes has a higher coreness score.

**Table 2 ijerph-19-06551-t002:** Multiple linear regression results for the COVID-19 severity measure.

	Delta Variant (from [15])			Omicron Variant (This Study)		
	Coefficient	*t*-Value	*p*-Value	Coefficient	*t*-Value	*p*-Value
Constant	−80.131	−0.970	0.334	6.979	8.50	0.000
Coreness	−0.098	0.000	1.000	−4.548	−1.014	0.313
Degree	1.46 × 10^5^	7.342	0.000	311.898	1.575	0.118
Closeness	−5.25 × 10^4^	−0.452	0.652	−968.561	−0.840	0.402
Betweenness	−1423.047	−0.819	0.415	−20.138	−1.165	0.246
Eigenvector	−1379.437	−3.944	0.000	−7.013	−2.017	0.046

**Table 3 ijerph-19-06551-t003:** Comparison of R-squared values across models using Delta and Omicron data. The R-squared values for the Delta variant were taken from Uddin et al. [15]. Since none of the 95% confidence interval (CI) range values include 0, they are statistically significant at *p* ≤ 0.05.

	$R^{2}$	*k*	*n*	$SE_{R_{i}^{2}}^{2}$	$SE_{R_{i}^{2}-R_{j}^{2}}$	95% CI	Sig.
Multiple linear regression							
$\mathrm{Delta}\;\left(MLR_{D}\right)$	0.358	5	130	0.0040	0.071	0.318 ± 0.142	≤0.05
$\mathrm{Omicron}\;\left(MLR_{O}\right)$	0.040	5	130	0.0010			
Random forest regression							
$\mathrm{Delta}\;\left(RFR_{D}\right)$	0.915	5	130	0.0002	0.065	0.525 ± 0.130	≤0.05
$\mathrm{Omicron}\left(RFR_{O}\right)$	0.363	5	130	0.0040			

## Data Availability

This study obtained research data from two publicly available sources: NSW Government’s COVID-19 data and statistics, and online map services (Google Maps and OpenStreetMap).

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
