# Peer review of "Comparing the Impact of Road Networks on COVID-19 Severity between Delta and Omicron Variants: A Study Based on Greater Sydney (Australia) Suburbs"

_ijerph, 2022, doi:10.3390/ijerph19116551_

Round 1

Reviewer 1 Report

Introduction. I understand that an article should start with a motivation and preferably a relatable example. But the numbers and citations mentioned in the first paragraph are already outdated. As the pandemic is a cataclysmic event that has affected the whole world like nothing else since World War Two, I believe that we can drop the conventions here and jump right to the last sentence of the first paragraph. I would suggest to then skip everything up to line 77 and continue the introductory paragraph with "Recently, a road network-based approach.." as the second sentence.
Construction of the network. The network construction described in lines 111-119 is utterly unsatisfactory. It may well be that after a lot of empirical research you conclude that you can abstract the local geography to such a primitive and small neworl. But in this day and age, why don't you use a comprehensive graph of all roads (what about suburban bus and rail service?) weighted by real traffic counts or at least by some Census-based commute data?
Results. "Since none of the 95% confidence interval (CI) range values includes 0, they are statistically significant at p ≤ 0.05". I have no idea what the authors are saying here. Based on Figure 1, I guess we can disregard the Closeness and Coreness measures. I am curious to learn how the authors interpret the other three and the respective different weights. In general, looking at Table 3, the considerable higher r-square for Delta makes perfect sense as Omicron is so infective that its spread is not a function of traveling distances but the mere sociability of the infected.
Discussion. I disagree with the observation that "The different number of road connections with neighbouring suburbs make the network features nonuniform. As a result, the non-linear property of the random forest algorithm better captured the variability and had better performance." But by raising this point, the authors also put their finger on a weakness of their approach. There is are techniques of spatial regression (rather than simple multiple regression) that deal with the notion of spatial autocorrelation - a factor completely neglected in the line of research reported here. The authors should have a look at https://gistbok.ucgis.org/bok-topics/spatial-autocorrelation and https://www.worldtransitresearch.info/research/6498/.
The discussion of the role of eigenvector centrality for the Omicron variant is interesting but hampered by the simplicity of the network. I am not convinced that the graph as constructed here bears much resemblance with the actual flows/interactions of people. Ironically, this discussion, using the lack of local travel restrictions for support flies in the face of the extra-ordinary transmissibility of the Omicron variant, which would speak for higher local intensities and much smaller kernel distribution sizes than the eigenvector method suggests.
One concern of mine is the heavy reliance on and similarity to Uddin [15]; I have sincere doubts that what is presented here warrants a new publication.

Author Response

Introduction. I understand that an article should start with motivation and preferably a relatable example. But the numbers and citations mentioned in the first paragraph are already outdated. As the pandemic is a cataclysmic event that has affected the whole world like nothing else since World War Two, I believe that we can drop the conventions here and jump right to the last sentence of the first paragraph. I would suggest skipping everything up to line 77 and continuing the introductory paragraph with "Recently, a road network-based approach.." as the second sentence.

Response: Thank you for this comment. Yes, some of the reported infection statistics have become outdated. We corrected them with an appropriate reference. Please see lines 31-44 for further details. We organised the introduction section as per the suggestion. Please see the introduction section of the revised manuscript.

Construction of the network. The network construction described in lines 111-119 is utterly unsatisfactory. It may well be that after a lot of empirical research, you conclude that you can abstract the local geography to such a primitive and small network. But in this day and age, why don't you use a comprehensive graph of all roads (what about suburban bus and rail service?) weighted by real traffic counts or at least by some Census-based commute data?

Response: Such a comprehensive graph can be constructed, and we also have access to the relevant. However, this study used pandemic severity data. Therefore, we considered the road network that would represent the human mobility options during a pandemic period. We thus took only the road connections in constructing the suburban road network. Although such a network can be based on all transportation media (e.g., road, bus and rail), this study considered only the road network due to the imposed lockdown on bus and rail services during the pandemic. Please see lines 122-127 for further details. The complete suburban network for the whole study region is presented in Figure 2.

Results. "Since none of the 95% confidence interval (CI) range values includes 0, they are statistically significant at p ≤ 0.05". I have no idea what the authors are saying here. Based on Figure 1, I guess we can disregard the Closeness and Coreness measures. I am curious to learn how the authors interpret the other three and the respective different weights. In general, looking at Table 3, the considerable higher r-square for Delta makes perfect sense as Omicron is so infective that its spread is not a function of travelling distances but the mere sociability of the infected.

Response: In Section 2.5 (Comparing R-squared values), we describe how to compare two different R-squared values using the underlying confidence interval. The two formulas in this section explain how to make this comparison. To make it clearer, we added additional text in the revised submission. If the 95% CI includes 0, there is no statistically significant difference between the underlying regression models. An inclusion of 0 in the 95% CI indicates a chance that the confidence could be 0. Otherwise, there is a statistically significant difference between them. Please see lines 180-181 for further details.

           Figure 1 is just an illustration of the entire suburban road network. The entire suburban road network can be found in Supplementary Figure 1. This aggregated network is much larger where any sort of network measures can be quantified. All of our network results are based on this network.

Discussion. I disagree with the observation that "The different number of road connections with neighbouring suburbs make the network features nonuniform. As a result, the non-linear property of the random forest algorithm better captured the variability and had better performance." But by raising this point, the authors also put their finger on a weakness of their approach. There are techniques of spatial regression (rather than simple multiple regression) that deal with the notion of spatial autocorrelation - a factor completely neglected in the line of the research reported here. The authors should have a look at https://gistbok.ucgis.org/bok-topics/spatial-autocorrelation and https://www.worldtransitresearch.info/research/6498/.
The discussion of the role of eigenvector centrality for the Omicron variant is interesting but hampered by the simplicity of the network. I am not convinced that the graph as constructed here bears much resemblance with the actual flows/interactions of people. Ironically, this discussion, using the lack of local travel restrictions for support flies in the face of the extra-ordinary transmissibility of the Omicron variant, which would speak for higher local intensities and much smaller kernel distribution sizes than the eigenvector method suggests.

One concern of mine is the heavy reliance on and similarity to Uddin [15]; I have sincere doubts that what is presented here warrants a new publication.

Response: Thank you for this comment. Yes, we could analyse the data using spatial regression. However, we used graph analytics to convert spatial data into a network/graph structure. We then applied complex network measures to quantify various aspects of that network. Finally, we applied multiple linear regression and compared the R-squared values using the approach described in Section 2.5 (Comparing R-squared values).

           Yes, Uddin [15] is the base article for this research. This article developed a network approach to investigate the impact of the suburban road network on COVID-19 severity and vulnerability only for the Delta variant. Some results of Tables 2 and 3 are also taken from this study. We compared the same results with the Omicron variant in the present study to explore the difference in spread pattern between the delta and omicron variants. We explicitly mentioned it in the introduction section. Please see lines 89-92 for further details.

Reviewer 2 Report

Dear authors,

First of all, I congratulate you on your courageous proposal to assess the effects of the various variants. However, in order to improve your proposal, I would recommend a cartographic contribution for the case studies developed in GIS. This contribution should be reflected in the results as well as in the discussion and conclusions of the article to provide a greater spatial vision of the phenomenon.

Without further, receive a best regards.

Author Response

First of all, I congratulate you on your courageous proposal to assess the effects of the various variants. However, in order to improve your proposal, I would recommend a cartographic contribution for the case studies developed in GIS. This contribution should be reflected in the results as well as in the discussion and conclusions of the article to provide a greater spatial vision of the phenomenon.

Response: Thank you for this very insightful comment. We added a new figure (Figure 2) to illustrate the entire road network for the study site. We also made the relevant discussion in the discussion section.

Reviewer 3 Report

This study presents an interesting topic. It explores the effect of suburban road network measures on COVID-19 severity between Delta and Omicron variants.  The method is reasonably sound and data are valid. However, revisions must be made in the following aspects. 

  • The title must be specified to suburban road network in the study area.
  • Introduction: More detailed background and problem statements should be added especially the relationship between road network configurations/characteristics and COVID-19 transmission. 
  • Methodology: This study compares between Delta and Omicron variant data by using different time frames (5-week for Delta and only 1-week for Omicron). Discuss this issue and recommend the expected results if different time frame were considered.
  • Consider illustrate the time frame, the statistics, the restriction, and travel patterns during Delta and Omicron  transmission in the study area. 
  • The road network used is rather small. It consists of 9 nodes connecting among suburbs. Discuss the impact of the size of network on the network analysis measures. 

Author Response

The title must be specified to the suburban road network in the study area.

Response: Thank you for this comment. We changed the title accordingly.

Introduction: More detailed background and problem statements should be added, especially the relationship between road network configurations/characteristics and COVID-19 transmission. 

Response: We added a more detailed background in the introduction section. Please see this section.

Methodology:

(a)This study compares Delta and Omicron variant data by using different time frames (5-week for Delta and only 1-week for Omicron). Discuss this issue and recommend the expected results if different time frames were considered.

(b)Consider illustrating the time frame, the statistics, the restriction, and travel patterns during Delta and Omicron transmission in the study area.

(c) The road network used is rather small. It consists of 9 nodes connecting among suburbs. Discuss the impact of the size of the network on the network analysis measures.

Response:

(a) We discussed this issue in the discussion section and made the relevant recommendation as suggested. Please see the discussion section, lines 292-297. (b) We mentioned it in the discussion section. (c) Figure 1 is just a sample of the entire road network. In this revised submission, we presented the entire road network in a new figure (Figure 2).